# Satisfaction and health-related quality of life of patients with microtia following reconstructive surgery using the Nagata technique

**Dini Widiarni Widodo** [1]*, **Robert Mars**[1], **Ronny Suwento**[1], **Widayat Alviandi**[1], **Imelda Ika Dian Oriza**[2], **Saptawati Bardosono**[3]

1 Department of Otorhinolaryngology, Head and Neck Surgery, Faculty of Medicine, Dr. Cipto Mangunkusumo Hospital, Universitas Indonesia, Jakarta, Indonesia, 2 Faculty of Psychology, Universitas Indonesia, Jakarta, Indonesia, 3 Department of Nutrition, Faculty of Medicine, Dr. Cipto Mangunkusumo Hospital, Universitas Indonesia, Jakarta, Indonesia

* dini_pancho@yahoo.com

**Data Availability Statement:** All relevant data is within the paper.

**Funding:** This study is partially granted by Universitas Indonesia with grant number NKB-

## Abstract

### Objective

This study aimed to investigate the functional outcomes, satisfaction, and quality of life of patients with microtia following reconstructive surgery.

### Methods

This cross-sectional study was conducted using retrospective data of patients with microtia following reconstructive surgery using the Nagata technique. Data were obtained from the medical records of patients who underwent reconstructive surgery at the Division of Facial Plastic and Reconstructive Surgery, Department of Otorhinolaryngology, Head and Neck Surgery, Dr. Cipto Mangunkusumo Hospital between 2014 and 2018. All eligible patients were referred to participate in this study between November 2018 and March 2019. The hearing function was assessed by a professional audiologist after surgery. Patient satisfaction was evaluated by interview using a previously developed questionnaire, while quality of life was assessed using the EuroQol-5D-Young questionnaire.

### Results

Thirty-one eligible subjects were included in the study. Pain and discomfort were the most commonly reported factors related to the quality of life following surgery. Approximately 67.7% of the patients were satisfied; 19.4% were very satisfied, and 12.9% reported acceptance of their surgical outcomes. The most common complication was infection (n = 8). Most patients did not experience any problems in their lives after microtia surgery.

### Conclusions

The highest rate of satisfactory outcomes was observed for the lobule subunit, which was assumed to be associated with the use of the Z-plasty technique. The most common

0370/UN2/R3.1/HKP.05.00/2019. The author who received this award is Dini Widiarni Widodo (DWW). The funders had no role in study design, data collection and analysis, decision to publish, or preparation of the manuscript.There was no additional external funding received for this study.

**Competing interests:** The authors have declared that no competing interests exist.

complication was infection, as environmental hygiene was the most important factor. Thus, further concern for maintaining good hygiene is necessary to improve the quality of reconstructive surgery. The level of satisfaction with microtia reconstructive surgery was adequate. Most patients had a good health-related quality of life without experiencing any problems.

## Introduction

The auricle is a part of the auditory organ, that transmits sound waves into the acoustic canal. It also plays an important role in an individual's appearance [1]. In clinical practice of Facial Plastic and Reconstructive surgery, microtia is the most common congenital ear abnormality observed [2]. Microtia is a congenital deformity in which the auricle fails to develop or does not develop properly. During embryogenesis, the auricles are derived from the first and second branchial arches. The branchial arches transform into six auditory tissue elevations (hillock of His). The elevations then merge into the entire ear. The absence of two to five elevations results in common and typical microtia. Ear malformations may occur at any stage during development [3].

Schloss estimates that the prevalence of microtia is more than 3:10,000 [4]. Microtia is more common in men than in women [5]. Approximately 70%-90% of affected individuals have a unilateral involvement. Further, microtia may develop as an isolated condition, as part of the anomaly spectrum, or as a syndrome, such as vertebral anomaly [6].

Patients with microtia frequently experience stigmatization from those around them, which may induce a negative self-perception of their competence and attractiveness and consequently affect their quality of life [7]. Moreover, hearing loss in patients with microtia may also have negative emotional impacts [8]. Awan et al. [9], found a significant reduction in life satisfaction and quality of life in adolescents living with microtia. Reconstructive surgery of microtia was found to reverse psychological distress and significantly improve patients' quality of life.

For reconstructive plastic surgery cases, patient perceptions and evaluations of quality of life are important indicators for assessment [10–12]. We used the EuroQol-5D-Young (EQ-5D-Y) questionnaire, which is a version of the EQ-5D questionnaire designed for children and teenagers, to evaluate quality of life [13, 14]. Another questionnaire used to evaluate satisfaction and perceptions regarding outer ear reconstructive surgery was the patient perception and satisfaction questionnaire for microtia reconstruction, which was developed by Cui et al. [15]. These questionnaires were used to describe the profiles of patients with microtia, especially satisfaction rates and quality of life after reconstructive surgery. The psychometric analysis demonstrated that the questionnaire met the criteria for Cronbach's $\alpha$ of $> 0.80$, internal consistency r of $>0.40$, test-retest reliability with r of $>0.7$, p-value of $<0.05$, interim correlation of $<0.8$, and interscale correlation of $<0.7$ [16].

This study is a novel study on patients with microtia in Indonesia, which may lead to a better understanding of patients' quality of life and experiences in reconstructive surgery.

## Materials and methods

Using data of patients who underwent microtia reconstructive surgery, this cross-sectional study followed up on the patients' quality of life and associated factors following reconstructive surgery. The study was conducted at the Division of Facial Plastic and Reconstructive Surgery,

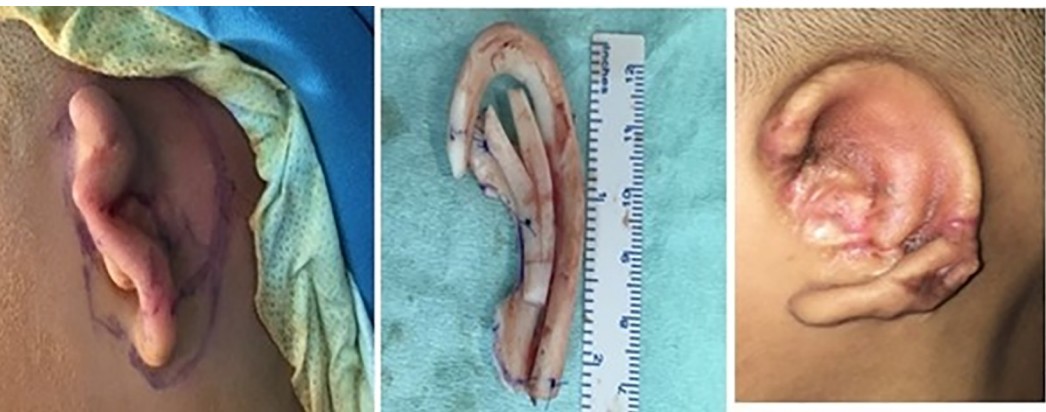

**Fig 1.** (a) Grade III malformation on the left ear of a 10-year-old girl diagnosed with bilateral microtia; (b) cartilage framework; and (c) reconstructed auricle at 6 months after surgery.

Department of Otorhinolaryngology, Head and Neck Surgery, Dr. Cipto Mangunkusumo Hospital, Jakarta, Indonesia. It was approved by The Ethics Committee of the Faculty of Medicine, Universitas Indonesia (No. 18-10-1237), and all participants provided written informed consent to share their evaluation findings after microtia reconstructive surgery using two questionnaires. Consent was obtained from the parents or guardians of participants under school age.

The total sampling method was used in this study. There were 40 patients with microtia who had undergone reconstructive surgery at the Division of Facial Plastic and Reconstructive Surgery, Department of Otorhinolaryngology, Head and Neck Surgery, Dr. Cipto Mangunkusumo Hospital between 2014 and 2018. Subjects were selected in accordance with the following inclusion criteria: (1) ability to speak the Indonesian language; (2) ability to read and write; and (3) willingness to participate in direct interviews. All patients were expected to come to the clinic for follow-up; however, if they were unable to come, the patients were called via phone to be asked if they were willing to conduct interviews at home. Meanwhile, subjects were excluded in accordance with the following exclusion criteria: (1) non-cooperation during interviews; (2) severe visual impairment; and (3) inability to understand the questionnaire. Thirty-one subjects met the inclusion criteria. All subjects were recruited by a single surgeon without any involvement of other surgeons to avoid bias from additional surgeons.

The subjects were patients with microtia who underwent two-step microtia surgery using the Nagata technique, that is, creating a framework in the first stage and elevation of the framework with or without atresiaplasty in the second stage, according the indication criteria of the Jahrsdoerfer score (Fig 1). The material for the framework was obtained from the seventh, eighth, and ninth rib cartilages, and a skin graft was harvested from the thigh for the elevation in the second stage [17–19]. The follow-up duration was at least 3 months after surgery, until all required data were obtained.

The postoperative data of the patients with microtia were collected from medical records, which included history of surgery, audiometry before and after surgery, computerized tomography scans of the mastoid bones, Jahrsdoerfer score, and complications. The patients were invited to participate in this study between November 2018 and March 2019. All patients were assessed via direct interviews by investigators who tracked the medical record data of all patients who underwent prior surgery. Hearing tests were performed by a skilled audiologist. The subjects were interviewed by a trained research assistant. They then filled out the

questionnaire by themselves, and if they found some difficulties, their parents could provide them guidance.

The instruments used to evaluate the patients in the study were the EQ-5D-Y questionnaire that was validated for the Indonesian language by the EuroQol group and the patient perception and satisfaction questionnaire for microtia reconstruction translated into the Indonesian language by the International Language Center, University of Indonesia. In our opinion, the Indonesian version of this questionnaire does not require further validation because the questions are relatively simple and arranged systematically without causing a misconception of language.

## Results

There were 8 male and 23 female subjects (25.8% vs. 74.2%) in our study. The mean age of participants was 13 years, with a standard deviation of 3.67 years. The youngest subject's age was 6 years old, and the oldest was 17 years old. The educational levels of the subjects were elementary school and high school graduates.

Third-degree microtia lobule types were the most common finding in our study (Table 1). The median Jahrsdoerfer score for the right side was 7, while that for the left side was 6. There were 20 microtia ears with a Jahrsdoerfer score of $\geq 7$ and 18 ears with a score of $< 7$; and one ear had an inconclusive score. The most common type of ear canal, both on the right and left sides, was ear canal atresia.

In this study, 24 subjects underwent pure tone audiometric examination, and seven underwent BERA examination. The type of hearing loss before surgery as examined via pure tone audiometry among 48 ears was conductive hearing loss in 27 ears, normal hearing in 15 ears, sensorineural hearing loss in 3 ears, and mixed hearing loss in 3 ears. The type of hearing loss after surgery was conductive hearing loss in 26 ears, normal hearing in 16 ears, sensorineural

**Table 1. Severity and type of microtia, Jahrsdoerfer score based on computed tomography results, and type of ear canal in the patients with microtia.**

| Severity (Ear unit) | Right ear | Left ear |
|---|---|---|
| | N = 62 ears | |
| **Normal ear lobe** | 11 | 12 |
| **First-degree microtia** | 0 | 1 |
| **Second-degree microtia** | 3 | 0 |
| **Third-degree microtia** | 17 | 18 |
| **Type** | | |
| Concha type | 3 | 3 |
| Sausage-shaped type | 1 | 3 |
| Lobule type | 13 | 12 |
| **Jahrsdoerfer score, median (min-max)** | 7 (2–9) | 6 (3–9) |
| **Jahrsdoerfer score** | | |
| $\geq 7$ | 20 | |
| $< 7$ | 18 | |
| Inconclusive | 1 | |
| Normal ear | 23 | |
| **Type of ear canal** | **Right ear** | **Left ear** |
| Normal | 14 | 10 |
| Stenosis | 0 | 3 |
| Atresia | 17 | 18 |

**Table 2. Audiometry results of the patients with microtia.**

| Mean hearing threshold | Before surgery (dB) | After surgery (dB) |
|---|---|---|
| **Pure tone audiometry** | | |
| Mean hearing threshold | 64.44 ± 17.52 | 62 ± 8.165 |
| Mean hearing threshold difference | 2.33 ± 5.55 | |
| **BERA: bone conduction** | | |
| Median (min-max) | 20 (20–65) | |
| **BERA: air conduction** | | |
| **Tone burst stimulation** | | |
| Median (min-max) | 60 (20–60) | |
| **BERA: air conduction** | | |
| **Click stimulation** | | |
| Mean | 67 ± 3.74 | |

hearing loss in 3 ears, and mixed hearing loss in 3 ears. The mean hearing threshold before surgery as examined via pure tone audiometry among 33 microtia ears was 64.44 ± 17.52 dB. The mean postoperative hearing threshold was 62 ± 8.165 dB. The mean difference in the hearing threshold before and after surgery was 2.33 ± 5.55 dB. In the bone conduction BERA examination, the median V-wave detection threshold was 20 (range, 20–65) dB. In the air conduction BERA examination of tone burst stimulation, the median V-wave detection threshold was 60 (range, 20–60) dB. In the air conduction BERA examination of click stimulation, the mean V-wave detection threshold was 67 ± 3.74 dB (Table 2).

The level of satisfaction with the surgical outcomes is shown in Fig 2. Approximately 67.7% of the patients were satisfied; 19.4% were very satisfied; and 12.9% reported that they could accept their surgical outcomes (Fig 2).

Regarding patient expectations, 71% of the patients were satisfied with their helix, while 74% were very satisfied with their lobules. Regarding the satisfaction level of "acceptable," 48.4% were satisfied with their triangular fossa. The patients were dissatisfied or very dissatisfied with their antihelix, scapha and concha, as these were not in accordance with their expectations (Fig 3).

The most common complication found in our study was infection (n = 8). The infection may cause framework cartilage exposure (n = 4), auricle subunit defect (n = 3), and cartilage resorption (n = 2). Meanwhile, the healing process of the wound caused ear canal stenosis (n = 6), keloid (n = 1), and posterior auricle synechia (n = 1), which required further revision surgery.

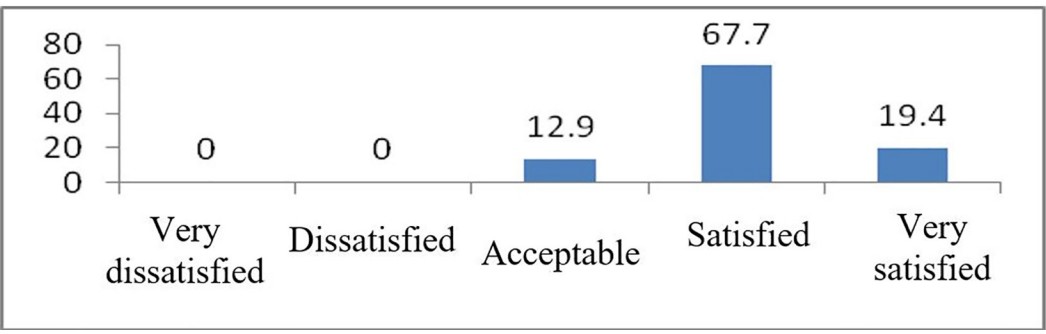

**Fig 2. Level of satisfaction with the surgical outcomes of microtia reconstructive surgery.**

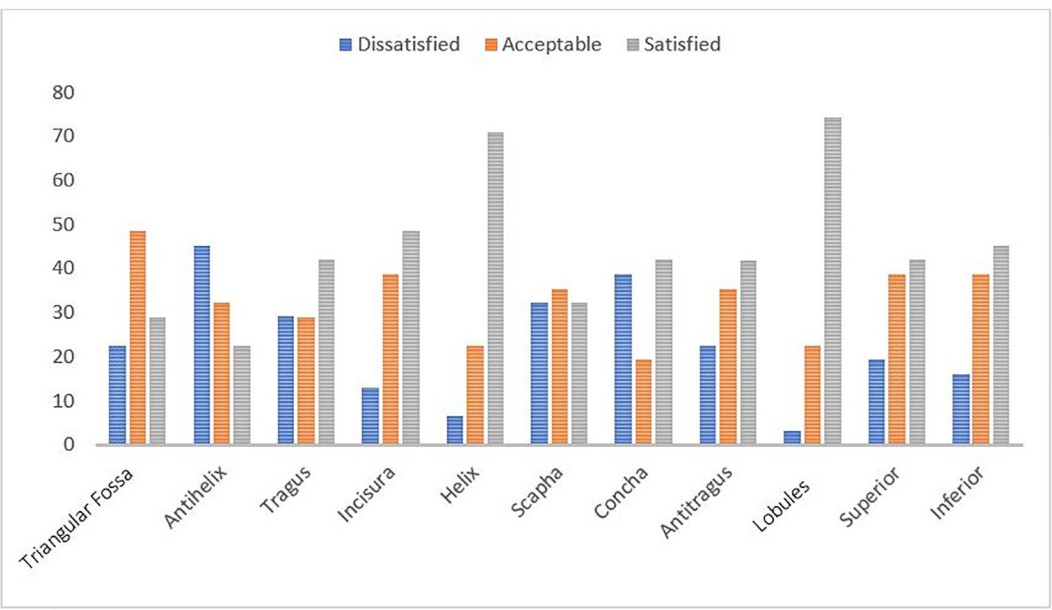

**Fig 3. Satisfaction level of the patients with microtia based on the aesthetic quality of the ear subunit.**

The factors associated with the postoperative quality of life of the patients with microtia are shown in Table 3. There were 3 patients who had moderate problems with mobility; 5 patients who experienced difficulty in looking after themselves; 12 patients who had pain or discomfort; 10 patients who still felt worried, sad, or unhappy; and 1 patient who was severely struggling to perform usual activities.

## Discussion

In our study, we found a greater number of female patients with microtia. This finding is different from the previous finding reported by Cui et al. [15] and Patil et al. [20], that is the majority of patients with microtia were male. This may have occurred in our study, as women are more likely to pay greater attention to their physical appearance than men [15, 20].

The most common microtia types in this study, both on the right and left side, were lobule-type and third-degree microtia. These findings are consistent with those reported by van Nunen et al. [21], who also found that third-degree and lobule-type microtia are the most common type in patients.

**Table 3. Factors associated with the postoperative quality of life of the patients with microtia.**

| Dimension | Problems | | No problem |
|---|---|---|---|
| | Severe | Moderate | |
| **Mobility (walking about)** | 0 | 3 | 28 |
| **Looking after my-self** | 0 | 5 | 26 |
| **Doing usual activities (going to school or performing hobbies and sports)** | 1 | 6 | 24 |
| **Having pain or discomfort** | 0 | 12 | 19 |
| **Feeling worried, sad, or unhappy** | 0 | 10 | 21 |
| **Self-health perception** | | | |
| Median (min-max) | 85 (50–100) | | |

The median Jahrsdoerfer score for the right ear in our study was 7, while that for the left ear was 6. These findings are in the same range as that of the previous study conducted by Marpaung [22] in Dr. Cipto Mangunkusumo Hospital, where the mean Jahrsdoerfer score was 6 with a standard deviation of 2.7 [22].

The type, degree, and mean hearing threshold in the patients with microtia before surgery in this study were severe conductive hearing loss with a mean hearing threshold of 64.44 ± 17.52 dB. This result is consistent with that of the study of Nicholas et al. [23], who reported that the mean hearing threshold before surgery was 61.9 dB, as measured via pure tone audiometry. Their study described the function of hearing examination not only for preoperative hearing function description but also for prediction improvement of hearing function after atresiaplasty [23]. A review conducted by Ruhl and Kesser [24] also suggested that the hearing threshold in patients with unilateral microtia with ear canal atresia ranged from 50 to 65 dB [24].

The mean postoperative hearing threshold in this study was 62 ± 8.165 dB. The mean increase in the hearing threshold after surgery in this study was 2.33 ± 5.5 dB. Yellon [25] investigated 19 patients with microtia with congenital atresia before and after microtia reconstructive surgery and atresiaplasty and found that the mean postoperative hearing threshold was 37.5 dB. This result is very different because we found complications of ear canal stenosis in the patients who underwent atresiaplasty (six out of eight subjects).

In the bone conduction BERA examination, the median detection threshold of the V wave was 20 (range, 20–65) dB. This finding is in accordance with that of the research by Kaga and Tanaka [26], who found a detection threshold ranging from 5 to 40 dB [26]. In the air conduction BERA examination of click stimulation, the mean V-wave detection threshold was 67 ± 3.74 dB. This is in accordance with that of the research by Patil et al. [20] and Marpaung [22], who evaluated the hearing threshold using pure tone audiometry and BERA examination and obtained a result of 44.3 dB [20, 22].

The educational level of the subjects was associated with their age range. In this study, the youngest subject who underwent first-stage reconstructive surgery was 6 years old. The age at which a patient should undergo surgery varies on the basis of the technique used. At the age of 6 years, the rib cage was expected to adequately develop to allow for sufficient harvesting of the sixth to eight costal cartilage as the framework [27]. Moreover, patients with microtia may experience psychological problems, such as receiving negative feedback from their peers during school age; therefore, microtia surgery is expected to reduce the psychological stressor [9].

Most subjects were satisfied with the surgical outcome of their helix and lobule, which is consistent with the findings by Cui et al. [15], who demonstrated that the highest satisfaction level was also found for the helix area. The highest rate of very satisfactory outcomes was observed for the lobule subunit. It is assumed that the findings were associated with the most common type of microtia cases and approaches in our study, which were the lobule types and the Z-plasty technique, respectively. For the category of very unsatisfactory outcomes, the subjects selected the antihelix subunit and the entire inferior region. This is attributed to the difficulty in performing duplication for this part, as it has a complex and small shape and requires an adequate amount of cartilage to form an antihelix. It also requires the use of negative pressure, such as suction, to create the entire ear lobe. The inferior part is considered essential in providing the entire image of a patient's ear lobe [15].

As previously stated, the most common complication found in our study was infection. This is consistent with the findings by Long et al. [28], who focused on complications in patients who underwent microtia reconstructive surgery using autologous cartilage grafts and found that infection was the most common complication. However, a similar study conducted by Mandelbaum et al. [29] showed that the most common complication was framework

cartilage exposure, followed by infection. Post-reconstructive surgery infection is associated with poor wound healing and graft failure [30–33]. Conversely, Fu et al. [34] showed delayed wound healing as the most common complication found in patients with microtia after reconstructive surgery [34]. It is assumed that the factor affecting these results is environmental hygiene. Indonesia is a developing country, and the incidence of health care-associated infections (HAIs) in developing countries is relatively high, particularly for surgical site infections (SSIs). According to a report from the Ministry of Health, Republic of Indonesia, the prevalence of SSIs is higher in public hospitals (0.99%) than in special private hospitals (0.77%). This may be attributed to a lack of monitoring and prevention measures, limited equipment, and a high number of patients in the hospital [35]. Another factor that may affect the results is the difference in skin structure between Asian and Caucasian patients. Asian skin has the weakest barrier function and is more sensitive to exogenous chemical agents, probably because of the thinner layer of the stratum corneum and higher density of the eccrine glands [36, 37]. The number of SSI cases in other developing countries, such as China and India, is also higher. Fan et al. [38] showed that the incidence of SSIs is 4.5% in China, and according to the Center for Disease Control and Prevention and the National Healthcare Safety Network in India, the rate of SSIs is 4.2%. The incidence of SSIs incidence ranges from 1.3% to 2.9%. The main possible cause is that there is no applicable law for implementing an infection control program [39]. Moreover, poor environmental conditions, poor hygiene, insufficient infrastructure and equipment, a lack of healthcare professionals, dense populations, a lack of knowledge and implementation of basic measures for infection control, and inappropriate long-term use of antibiotics can also affect the rate of SSIs [40].

As a regulation of Dr. Cipto Mangunkusumo Hospital to prevent surgical wound infection in our study, intravenous cefazolin was administered as prophylactic antibiotic 30 minutes before surgery. This is similar with the study by Agustina et al. [41], who showed that the most commonly used antibiotic in ENT surgery is cefazolin (92.5%). In addition, the hospital regulation by the Ministry of Health of the Republic of Indonesia Number 2406/MENKES/PER/XII/2011 regarding the general guidelines for the use of antibiotics also stated that first- and second-generation cephalosporins are recommended as surgical prophylaxis [41, 42]. According to Koento and Saleh [43], the optimal time to provide a prophylactic antibiotic was 0–30 minutes before the incision. The risk of infection would increase if prophylactic antibiotics are administered > 30 minutes before or after the incision.

According to the American Academy of Otolaryngology–Head and Neck Surgery, definitive therapy recommendations are adjusted to disease-causing organisms; however, the most recommended are penicillin groups [44]. The recommendations for empirical therapy may vary depending on the disease; however, the majority recommended are beta-lactam groups (penicillins and cephalosporins). This indicates that the antibiotic administration in this study was in line with therapeutic recommendations because the majority of those used were ceftriaxone, which belongs to the beta-lactam group.

Wound infection occurred at discharge and 1month post-surgery; the post-surgical care of the wound at home was poor, and the patients would usually manipulate the wound. Indonesia is a tropical country with a hot humid climate, with an average humidity level of 70%–90% for most of the year, which can cause excessive sweat production and itchiness to the patients' skin. As previously stated, the thin dermal layer of the auricle made this part more prone to SSIs [36, 37].

The strength of our study was that it is the first study of its kind in Indonesia. The study also used the EQ-5D-Y questionnaire, for which direct official translation was provided by the EuroQol group and was validated in Indonesia. Meanwhile, a limitation of this study was that it used a cross-sectional design. The patients' quality of life was evaluated only once; therefore,

the data on quality of life may have changed according to subjective factors. The use of different questionnaires may also be one of the factors that affected the study results. The number of subjects who fulfilled the study criteria was relatively small compared to that in previous studies [29, 45]. The small sample size obtained from the statistical calculations was also a concern. The power of the study—achieved with a sample size of 31 subjects was 80%. However, considering that the protocol of microtia surgery in our study is still relatively consistent, the results of our study can still be applied and studied further.

## Conclusion

In general, the level of satisfaction with microtia reconstructive surgery in Indonesia is adequate. Most patients had a good health-related quality of life without experiencing any problems after surgery. Infection after surgery should be the main concern for improving the quality of reconstructive surgery.

## Author Contributions

**Conceptualization:** Dini Widiarni Widodo, Robert Mars.

**Data curation:** Robert Mars.

**Formal analysis:** Robert Mars, Widayat Alviandi, Imelda Ika Dian Oriza.

**Investigation:** Ronny Suwento.

**Software:** Widayat Alviandi.

**Supervision:** Dini Widiarni Widodo, Imelda Ika Dian Oriza, Saptawati Bardosono.

**Validation:** Ronny Suwento.

**Writing – original draft:** Dini Widiarni Widodo.

**Writing – review & editing:** Dini Widiarni Widodo.

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
