## [Decision Letter · Decision Letter 0]

13 May 2021

PONE-D-21-07019

Satisfaction and health-related quality of life of microtia patients following reconstructive surgery using the Nagata technique

PLOS ONE

Dear Dr. Widodo,

Thank you for submitting your manuscript to PLOS ONE. After careful consideration, we feel that it has merit but does not fully meet PLOS ONE’s publication criteria as it currently stands. Therefore, we invite you to submit a revised version of the manuscript that addresses the points raised during the review process.

We look forward to receiving your revised manuscript.

Kind regards,

Jorge Spratley, MD, PhD

Academic Editor

PLOS ONE

Journal Requirements:

2. To comply with PLOS ONE submission guidelines, in your Methods section, please provide additional information regarding your statistical analyses. For more information on PLOS ONE' expectations for statistical reporting, please see https://journals.plos.org/plosone/s/submission-guidelines.#loc-statistical-reporting.

4. Please amend the manuscript submission data (via Edit Submission) to include authors Robert Mars, Ronny Suwento, Widayat Alviandi, Imelda Ika Dian Oriza, Saptawati Bardosono.

5. Please ensure that you refer to Figure 2 in your text as, if accepted, production will need this reference to link the reader to the figure.

Reviewers' comments:

Reviewer's Responses to Questions

**Comments to the Author**

1. Is the manuscript technically sound, and do the data support the conclusions?

Reviewer #1: Partly

2. Has the statistical analysis been performed appropriately and rigorously? 

Reviewer #1: I Don't Know

3. Have the authors made all data underlying the findings in their manuscript fully available?

Reviewer #1: Yes

4. Is the manuscript presented in an intelligible fashion and written in standard English?

Reviewer #1: No

5. Review Comments to the Author

Reviewer #1: There are several aspects which should be improved by the authors:

- The English language is not clear and correct. Some examples: ”Facial Plasty and Reconstructive(…)”, “This is caused by the difficulty in performing duplication(…)”

- The data of the study includes 31 patients which underwent surgery between 2014 and 2018. What was the total number of patients submitted to surgery? How were the patients enrolled? There are no patients lost to follow-up after surgery? All of them were interviewed? How was the interview conducted? By phone?

- Is the “Patient Perception and Satisfaction Questionnaire for Microtia Reconstruction” validated for Indonesian language?

- Where the questionnaires filled by parents or the patients themselves since some of them were adolescents?

- What was the follow-up time of the patients?

- “The education levels of subjects were elementary school and high school graduates.”- these represent a wide range of education levels. It should be further explained.

- How was the power of the study “re-calculated”, as stated in results?

- In methods authors describe “hearing tests” and “audiometry”, but they don’t mention any audiometric results.

- References should be more precise and correct. For example, in page 4 authors cited references number 9 and 10 of bibliography as examples of quality of life evaluations after reconstruction surgery. However, papers 9 and 10 evaluate otoplasty and prosthetic auricular reconstruction.

- Is there a correlation between the number of infections and the type of deformity or the surgical procedure characteritics?

- The discussion focus a lot on the infection and environmental hygiene in developing countries. However, for a correct evaluation of infections it should be described the protocol of antibiotic therapy, the number of days before being discharged after surgery and several other factors that can influence the infections rate.

- At least one recent and very relevant paper was not cited and considered by the authors: Patient Perception and Satisfaction Questionnaire for Microtia Reconstruction: A New Clinical Tool to Improve Patient Outcome- Cui et al 2018.

6. PLOS authors have the option to publish the peer review history of their article (what does this mean?). If published, this will include your full peer review and any attached files.

Reviewer #1: No

---

## [Author Response · Author response to Decision Letter 0]

10 Jul 2021

Dear Jorge Spratley, MD, PhD

Thank you for giving me the opportunity to submit a revised version of my manuscript titled “Satisfaction and health-related quality of life of microtia patients following reconstructive surgery using the Nagata technique” to PLOS ONE. 

I appreciate the time and effort that you and the reviewer have dedicated to provide your valuable feedback on my manuscript. I am grateful to the reviewer for their insightful comments on my paper. I have been able to incorporate changes to reflect most of the suggestions provided by the reviewer. I have highlighted the revised parts with “track change” within the manuscript. 

Here is a point-by-point response to the reviewers’ comments and concerns.

Comments to the author

1. Is the manuscript technically sound, and do the data support the conclusions?

Reviewer #1: Partly

Author response: We hope that the explanation on method section can show that this study had been conducted rigorously. We try to explain the step by step of this study systematically.

2. Has the statistical analysis been performed appropriately and rigorously?

Reviewer #1: I Don't Know

Author response: Thankyou for the comment. The data in this study was recorded in a research status form. After editing and coding, the data is analyzed with the SPSS program (Statistical Package) for Social Sciences) version 20. The data were analyzed descriptively. Data with a numerical scale is presented in the form of mean and standard deviation if the data distribution is normal; and presented in the form of the median and the value of minimum-maximum if the data distribution is not normal. Meanwhile, a categorical data presented in frequency (n) and percentage (%).

3. Have the authors made all data underlying the findings in their manuscript fully available?

The PLOS Data policy requires authors to make all data underlying the findings described in their manuscript fully available without restriction, with rare exception (please refer to the Data Availability Statement in the manuscript PDF file). The data should be provided as part of the manuscript or its supporting information or deposited to a public repository. For example, in addition to summary statistics, the data points behind means, medians and variance measures should be available. If there are restrictions on publicly sharing data e.g. participant privacy or use of data from a third party those must be specified.

Reviewer #1: Yes

Author response: All the data presented within the manuscript are fully avalaible

4. Is the manuscript presented in an intelligible fashion and written in standard English?

Reviewer #1: No

Author response: We are extremely sorry for our bad English. In order to present a better English, we used the language editing service provided by editage.com and revised the manuscript content based on the language editor suggestion.

5. Reviewer comments to the author:

- Comment: The English language is not clear and correct. Some examples: ”Facial Plasty and Reconstructive(…)”, “This is caused by the difficulty in performing duplication(…)”

Response: We are very sorry regarding our bad English. In this revised version of manuscript, we use language editing service by editage.com. We already changed the manuscript based on language editor’s feedback.

- Comment: The data of the study includes 31 patients which underwent surgery between 2014 and 2018. What was the total number of patients submitted to surgery? How were the patients enrolled? There are no patients lost to follow-up after surgery? All of them were interviewed? How was the interview conducted? By phone?

Response: The sampling method used in this study was total sampling. There were a total of 40 patients who undergone microtia reconstruction surgery between January 2014 and December 2018. The patients participated in this study was selected based on inclusion criteria: 1) Patients could speak Indonesian language; (2) Patients could read and write; (3)Patients were willing to participate in a direct interviews and the exclusion criteria were (1) Patients were not cooperative during interviews; (2) Patients with severe visual impairment; (3) Patients were unable to understand the questionnaire. There were 31 subjects who met the inclusion criteria.

And then, investigator collected post-operative microtia patients’ data from medical record including the history of surgery, audiometry before and after the surgery, computerized tomography (CT) scans of the mastoid bones, Jahrsdoerfer scores and complications after surgery. 

The patients were called to participate in this study between November 2018 and March 2019. There are no patients lost to-follow up after surgery. The interviews were performed by direct interviewed. Phone call was only for communication to determine the appointment date of direct interview. 

- Comment: Is the “Patient Perception and Satisfaction Questionnaire for Microtia Reconstruction” validated for Indonesian language?

Response: The Patient Perception and Satisfaction Questionnaire for Microtia Reconstruction have never been validated for Indonesian language. The translation process for this questionnaire was performed by International Langue Center, University of Indonesia (LBI FIB UI). In our opinion, the Indonesian version of this questionnaire does not need a further validation because the questions are relatively simple and arranged systematically without causing a misconception of language.

- Comment: Where the questionnaires filled by parents or the patients themselves since some of them were adolescents?

Response: Most of the subjects who participated in this study were cooperative, they could fill out the questionnaires by themselves. There were only small number of subjects who was assisted by their parents in filling out the questionnaires.

- Comment: What was the follow-up time of the patients?

Response: All patients were followed-up at least 3 months after surgery (the time which we predicted that post-operative wound would heal and dry) and until all the required data was obtained. 

- Comment: “The education levels of subjects were elementary school and high school graduates.”- these represent a wide range of education levels. It should be further explained.

Response: The education level of the subjects was associated with the subjects age range. In this study, the youngest subject age underwent first stage reconstruction surgery was 6 years old. At the age of 6 years old, the rib cage was expected to adequately develop to allow for sufficient harvesting of 6th -8th costal cartilage as the framework. In addition, according to studies comparing the results of psychosocial functioning in patients Microtia who undergone otoplasty at various age ranges, the best age to perform the reconstruction surgery is before the age of 7 year. 

- Comment: How was the power of the study “re-calculated”, as stated in results?

Response: Based on the sampling formula, we calculated that the subject amount required in this study was 43 subjects. But there were only 40 patients who underwent reconstruction surgery between January 2014 and December 2018 and only 31 patients who met the inclusion criteria. So we made a re-calculation on the power of this study based on the amount of the subject (31 subjects), and we get the result of 80%.

- Comment: In methods authors describe “hearing tests” and “audiometry”, but they don’t mention any audiometric results.

Response: Thank you for the insightful comment. In this revised version we have add the data of audiometry as well as the discussion. In this study, 24 subjects underwent pure tone audiometric examination, and 7 subjects underwent BERA examination. The mean of hearing threshold examined with pure tone audiometry before surgery on 33 microtia ears was 64.44± 17.52 dB. Meanwhile, the mean postoperative hearing threshold was 62 ± 8.165 dB. The mean difference in hearing threshold before and after surgery was 2.33 ± 5.55 dB. In the bone conduction BERA examination, the median value of V-wave detection threshold was 20 (20 – 65) dB. In the air conduction BERA examination of tone burst stimulation, the median value of the V wave detection threshold was 60 (20 – 60) dB. In the click stimulation BERA examination, the mean value of the V wave detection threshold is 67 ±3.74 dB.

- Comment: References should be more precise and correct. For example, in page 4 authors cited references number 9 and 10 of bibliography as examples of quality of life evaluations after reconstruction surgery. However, papers 9 and 10 evaluate otoplasty and prosthetic auricular reconstruction.

Response: We have taken reviewer’s comment and we have removed the reference number 9 and 10, and replaced it with other study by Awan et al that studied the ear reconstruction using costal cartilage.

- Comment: Is there a correlation between the number of infections and the type of deformity or the surgical procedure characteristics?

Response: There is no correlation between infection and the type of deformity but there is a correlation between infection and post-operative care procedure which will be discussed in the next answer to reviewer about factors that can influence infection rate.

- Comment: The discussion focus a lot on the infection and environmental hygiene in developing countries. However, for a correct evaluation of infections it should be described the protocol of antibiotic therapy, the number of days before being discharged after surgery and several other factors that can influence the infections rate.

Response: In the manuscript we already explained other factors that influence SSIs rate such as Asian skin thinner layer and the humidity in a tropical country. We also add the discussion regarding the antibiotic we used to prevent the SSIs

- Comment: At least one recent and very relevant paper was not cited and considered by the authors: Patient Perception and Satisfaction Questionnaire for Microtia Reconstruction: A New Clinical Tool to Improve Patient Outcome- Cui et al 2018.

Response: In the revised version, we have cited the following study by Cui et al 2018 that stated : the psychometric analysis demonstrated that the questionnaire met the criteria for Cornbach’s α > 0.80; internal consistency r>0.40, test-retest reliability with r>0.7 and P <0.05; interim correlation <0.8 and interscale correlation <0.7

We look forward to hearing from you regarding our submission and to respond to any further questions and comments you may have.

Sincerely,

Dini Widiarni Widodo, M.D, ORL-HNS

---

## [Editor Report · Decision Letter 1]

12 Aug 2021

Satisfaction and health-related quality of life of patients with microtia following reconstructive surgery using the Nagata technique

PONE-D-21-07019R1

Dear Dr. Widodo,

We’re pleased to inform you that your manuscript has been judged scientifically suitable for publication and will be formally accepted for publication once it meets all outstanding technical requirements.

Kind regards,

Jorge Spratley, MD, PhD

Academic Editor

PLOS ONE

Additional Editor Comments (optional):

Congratulations. Your manuscript has now reached the level for publication at Plos One.
---

## [Editor Report · Acceptance letter]

24 Aug 2021

PONE-D-21-07019R1 

Satisfaction and health-related quality of life of patients with microtia following reconstructive surgery using the Nagata technique 

Dear Dr. Widodo:

I'm pleased to inform you that your manuscript has been deemed suitable for publication in PLOS ONE. Congratulations! Your manuscript is now with our production department. 

Kind regards, 

on behalf of

Professor Jorge Spratley 

Academic Editor

PLOS ONE